# Persistence of Anti-ZIKV-IgG over Time Is Not a Useful Congenital Infection Marker in Infants Born to ZIKV-Infected Mothers: The NATZIG Cohort

**DOI:** 10.3390/v13040711

**Published:** 2021-04-20

**Authors:** Conrado Milani Coutinho, Juliana Dias Crivelenti Pereira Fernandes, Aparecida Yulie Yamamoto, Silvia Fabiana Biason de Moura Negrini, Bento Vidal de Moura Negrini, Sara Reis Teixeira, Fabiana Rezende Amaral, Márcia Soares Freitas da Motta, Adriana Aparecida Tiraboschi Bárbaro, Davi Casale Aragon, Magelda Montoya, Eva Harris, Geraldo Duarte, Marisa Márcia Mussi-Pinhata

**Affiliations:** 1Department of Gynecology and Obstetrics, Ribeirão Preto Medical School, University of São Paulo, Ribeirão Preto 14049-900, SP, Brazil; gduarte@fmrp.usp.br; 2Department of Pediatrics, Ribeirão Preto Medical School, University of São Paulo, Ribeirão Preto 14049-900, SP, Brazil; judcp_53@yahoo.com.br (J.D.C.P.F.); yulie@fmrp.usp.br (A.Y.Y.); sissinegrini@gmail.com (S.F.B.d.M.N.); bnegrini@hcrp.usp.br (B.V.d.M.N.); biarezendeamaral@yahoo.com.br (F.R.A.); marciasfmotta@hotmail.com (M.S.F.d.M.); a.tiraboschi@usp.br (A.A.T.B.); dcaragon@fmrp.usp.br (D.C.A.); 3Department of Imaging, Hematology and Oncology, Ribeirão Preto Medical School, University of São Paulo, Ribeirão Preto 14049-900, SP, Brazil; reisteixeirasara@gmail.com; 4Division of Infectious Diseases and Vaccinology, School of Public Health, University of California, Berkeley, CA 94720-7360, USA; mageldamontoya@yahoo.es (M.M.); eharris@berkeley.edu (E.H.)

**Keywords:** Zika virus, congenital infection, congenital Zika syndrome, IgM antibodies, IgG antibodies, diagnosis, infant, serology

## Abstract

Confirming ZIKV congenital infection is challenging because viral RNA is infrequently detected. We compared the presence of anti-ZIKV-IgM and the persistence of anti-ZIKV-IgG antibodies over 18 months in two cohorts of infants born to ZIKV-infected mothers: Cohort one: 30 infants with typical microcephaly or major brain abnormalities (Congenital Zika Syndrome-CZS); Cohort two: 123 asymptomatic infants. Serum samples obtained within 6 months of age were tested for anti-ZIKV-IgM. Anti-ZIKV-IgG was quantified in sequential samples collected at birth, 3–6 weeks, 3, 6, 12, and 18 months. ZIKV-RNA was never detected postnatally. Anti-ZIKV-IgM antibodies were detected at least once in 15/25 (60.0%; 95%CI: 38.7–78.9) infants with CZS and in 2/115 (1.7%; 95%CI: 0.2–6.1) asymptomatic infants. Although anti-ZIKV-IgG was always positive within 3–6 weeks of age, IgG levels decreased similarly over time in both cohorts. IgG levels decreased similarly in ZIKV-IgM-positive and ZIKV-IgM-negative CZS infants. Differently from other congenital infections, IgM would fail to diagnose 40% of severely symptomatic infants, and the persistence of IgG is not a useful marker for discriminating congenital infection among infants exposed to maternal ZIKV infection.

## 1. Introduction

The most well-known consequences of the intrauterine transmission of Zika virus (ZIKV) infection from mother to fetus are microcephaly and related severe brain abnormalities with neurological effects, defined as Congenital Zika Syndrome (CZS) [1]. However, as with other congenital infections, the ZIKV might also have a broad clinical spectrum [2,3,4]. Overall, considering estimates of a 20–30% vertical transmission rate [4] and the reported estimates of pregnancy losses (4–7%) and congenital disabilities characterized by CZS (5–14%) [5], one could suppose that up to 10% of infants born to ZIKV-infected mothers would be infected but clinically inapparent, limiting the identification of the virus [2]. Since the primary targets of ZIKV infection are mainly the central nervous system and eyes, infected infants without severe end-organ disease are most likely to benefit from long-term follow-up and the prevention of developmental delays by stimulation and rehabilitation. The best scenario would be to define the infection as early as possible to focus follow-up efforts exclusively on the infected infants.

Usually, molecular methods that detect viral nucleic acids are best used to postnatally diagnose latent viral congenital infections, such as human immunodeficiency virus and cytomegalovirus [6]. However, the confirmation of congenital ZIKV infection in infants using molecular methods is challenging. Although it has not been thoroughly studied, available data demonstrate that viral RNA is infrequently and transitorily detected [7,8], even in infants for whom ZIKV-RNA was found during fetal life [9]. Alternatively, specific IgM or the persistence of IgG antibodies over time are used to define other congenital infections such as toxoplasmosis [10]. The utility of serological tests has not been systematically studied in infants perinatally exposed to ZIKV.

The main objective of this study was to compare the detection of anti-ZIKV-IgM within 6 months of age and the dynamics of anti-ZIKV-IgG antibodies over 18–24 months of age between infants with typical CZS and asymptomatic infants whose mothers had ZIKV infection during pregnancy. We compared the early detection of ZIKV-RNA and ZIKV-IgM antibodies between groups in a subset of infants.

## 2. Materials and Methods

### 2.1. Study Population

This study was embedded in the Natural History of Zika Virus Infection in Gestation (NATZIG) project, which is a prospective population-based cohort study undertaken in the Ribeirão Preto region, Northeastern São Paulo State, Brazil [2]. This region experienced an outbreak of ZIKV infection that peaked during the last trimester of 2015 and the first trimester of 2016. Overall, 1116 pregnant women had flavivirus-like symptoms in this region, and 511 were confirmed as having ZIKV infection [2]. All study procedures received ethical approval (Processes #7404/2016, 24 June 2016), and written informed consent was obtained from all participants.

### 2.2. Study Design

We composed two cohorts of infants for this analysis. Cohort one (CZS) included infants born in or referred to the HCFMRP-USP University Hospital (Hospital das Clínicas da Faculdade de Medicina de Ribeirão Preto da Universidade de São Paulo) diagnosed with microcephaly and ZIKV-like patterns of brain abnormalities detected at birth or within 6 months of age whose mothers had confirmed or suspected ZIKV infection during pregnancy, and no other etiology was defined for central nervous system findings. Cohort two (Asymptomatic): Infants born at HCFMRP-USP whose mothers had ZIKV infection confirmed via the detection of ZIKV-RNA by polymerase chain reaction (PCR) in the blood (within 5 days of symptoms) and/or urine (within 8 days) taken for RNA-ZIKV testing [11]. They were born with clinical neurologically normal exams and had no microcephaly or Zika-like features in brain imaging.

We evaluated infants at entry with a complete physical and neurological exam. Birth weight and head circumference were measured before hospital discharge and classified according to the INTERGROWTH-21st criterion [12]. The neonatal hearing screening was done using both Transient Otoacoustic Emissions (TOAE) and automated auditory brainstem evoked responses (AABR-35dBHL). A complete eye examination included indirect ophthalmoscopy. Cranial ultrasonography and/or other neuroimaging studies were performed within 3 months of age by an experienced physician and reviewed by two board-certified radiologists. All infants from the CZS-cohort had cranial ultrasonography showing a Zika-like pattern of brain abnormalities defined by the presence of malformations of cortical development, such as gyral simplification and loss of the gray matter–white matter junction, with or without subcortical calcifications or ventriculomegaly [1]. Additional imaging findings, such as calcifications, ventriculomegaly, ventricular adhesions or periventricular cavities, corpus callosum dysgenesis, abnormal echogenicity of the white matter, enlargement of cerebrospinal fluid (CSF) spaces, and signs of brain volume loss [13,14] could also be present. None of the asymptomatic infants had abnormal brain imaging findings potentially related to CZS, which means there were no signs of malformations in cortical development characteristic of the Zika-like pattern nor additional abnormalities. 

Both groups of infants were followed from their entry into the study. Study visits were planned to occur at birth, 3–6 weeks, 3 ± 1, 6 ± 1, 9 ± 1, 12 ± 1, 18 ± 3, and 24 ± 3 months of age. Anthropometry, general conditions, growth, neurodevelopment, feeding, and morbidity were recorded in all visits in which a blood sample was obtained for testing.

### 2.3. Laboratory Testing

Laboratory personnel performing the tests were blinded to the infants’ conditions. Serum samples obtained at birth or within 6 months of age were tested for anti-ZIKV-IgM antibodies (ZIKV *Detect^TM^* 2.0 IgM Capture ELISA, InBios International, Inc., Seattle, WA, USA) [15], which was repeated at later time points when positive. Anti-ZIKV-IgG was tested in sequentially obtained samples using a commercially available anti-Zika-IgG ELISA (Diagnostic Bioprobes Srl, Sesto San Giovanni, MI, Italy) [16]. ZIKV-RNA testing by real-time-reverse transcription polymerase chain reaction (RT-PCR) was assayed in blood and urine within 3 months of age by the Adolfo Lutz Institute [11]. A second antibody-based assay used to detect anti-ZIKV-IgG antibodies, the NS1 blockade-of-binding (BOB) ELISA assay [17], was applied in a subset of infants for whom remaining sera were available at 6 and 18 months of age.

### 2.4. Data Analysis

We adjusted a Bayesian linear regression model to estimate all the anti-ZIKV-IgG mean differences between Cohorts one and two and 95% credible intervals after an exploratory data analysis. Time until undetectable IgG was considered as the interval between the date of birth and the last positive measurement for IgG added to the midpoint between the latest positive and the first negative measurements. The Kaplan–Meier analysis was used to illustrate time to undetectable IgG over time between two cohorts and between infants with detectable or undetectable anti-ZIKV-IgM from cohort one, considering only those infants who had at least two IgG measurements. Survival curves were compared using the log-rank test. We used the R 4.0.2 (R Foundation for Statistical Computing, Vienna, Austria) software.

## 3. Results

Among 41 infants with microcephaly or abnormal brain imaging findings who were born in or referred to our hospital, 30 were selected for the CZS cohort. The remaining 11 were not included in the study for the following reasons: no availability of brain imaging or blood samples (4), no ZIKV-like cranial sonography pattern (4), congenital cytomegalovirus infection (2), and fetal alcohol syndrome (1). Of the 125 asymptomatic infants fulfilling the selection criteria, two mothers did not consent to enrollment. Table 1 shows the infants’ characteristics. As expected, there were higher frequencies of maternal infection in the first trimester of gestation (66.7% versus 21.9%) and preterm births (16.7% versus 8.7%) for the cohort one infants compared to those from cohort two. Similarly, birth weight (2694 g versus 3101 g) and head circumference (28.3 cm versus 33.4 cm) were consistently lower in CZS infants compared to asymptomatic ones. All infants classified as CZS had major detectable neurologic abnormalities. Information regarding gestational age of maternal infection and at delivery were not available only for two and three neonates, respectively.

### 3.1. ZIKV-RNA and Anti-ZIKV-IgM Antibodies

As shown in Table 2, about half of the infants in Cohort one and a third of those in Cohort two had samples submitted for ZIKV-RNA testing, all with negative results. There was just one indeterminate result for an infant from Cohort two, and for the remaining infants no results were available.

Anti-ZIKV-IgM antibody tests were performed on samples from 91.5% of the 153 participants, yielding positive results in 17/140 (12.1%). Among Cohort one, after the exclusion of five infants in which it was not possible to perform the test, anti-ZIKV-IgM antibodies were detected at least once in 15/25 (60.0%, 95%CI: 38.7–78.9) infants. These infants had a median of 4 (1–9) tested samples each. Overall, for those with an initial positive sample, anti-ZIKV-IgM reactivity in sequential samples was found in 8/13, 6/13, 3/6, 2/3, and 1/1 infants respectively at 3, 6, 12, 18, and 24 months of age. Interestingly, IgM was detected in two to three sequential specimens, usually up to 6–12 months of age, before turning negative. One infant, who had six tested samples, was positive until 18 months and negative at 24 months. We detected no fluctuating anti-ZIKV-IgM. After exclusion of the eight infants without any available IgM results, only two out of 115 (1.7%; 95%CI: 0.2–6.1) infants from Cohort two had positive test results. One of these was positive at birth, 3–6 weeks, and 3 months of age and negative at 6 months. The remaining infant was tested only at 3, 18, and 24 months. He was positive at 3 and 18 and negative at 24 months of age.

### 3.2. Anti-ZIKV-IgG Antibodies

Cohort one and cohort two infants had, respectively, a median of 4 (range: 1–6) and 3 (range: 1–6) specimens obtained from birth to 18 months of age for anti-ZIKV-IgG testing. In both cohorts, most (87.7%) infants had ≥2 tested serum samples. Figure 1A shows the IgG levels detected over time. There was a parallel decay in IgG median levels with advancing age in both groups, from 8.7 arbitrary units (IQR range: 6.7–10.1) at birth to 1.0 (0.3–1.9) at 6 months in Cohort one and from 9.0 (IQR range: 6.2–10.2) at birth to 0.3 (0.3–1.3) at 6 months in Cohort two. Notably, the median level of IgG at 6 months had decreased to 89% of levels detected at birth in Cohort one and 97% in Cohort two. Anti-ZIKV-IgG titers were similar at birth and 3–6 weeks. We detected a small difference at 3, 6, and 12 months of age (Figure 1B). Infants with CZS had slightly higher levels of ZIKV-IgG at these ages than the asymptomatic ones. However, it was statistically significant only at 6 and 12 months.

At six months of age, 42.3% of Cohort one and 58.1% of Cohort two infants had lost anti-ZIKV-IgG, while 92.3% and 99.1%, respectively, had undetectable IgG antibodies by 11 months of age. All infants from cohort one lost IgG by 15 months of age, and all from Cohort two had undetectable IgG by 12 months of age. The Kaplan–Meier analysis showed that the time to undetectable IgG was similar in both cohorts (*p* = 0.15) (Figure 2A).

We analyzed the IgG decline over time among the 25 infants with CZS for whom samples for anti-ZIKV-IgM were available to verify whether the IgG decay occurring in the IgM-positive infants (15) was slower than that occurring in the IgM-negative infants (10). However, they had a similar time to undetectable IgG, as shown by the Kaplan–Meier analysis (*p* = 0.59) (Figure 2B).

The detection of anti-ZIKV-IgG was even less frequent using the BOB assay, which was applied to a subset of infants with remaining samples at 6 and 18 months of age. Among 18 infants from the CZS cohort tested at 6 months of age, only one (5.6%) had a borderline result, while the remaining were negative. All of the 19 infants tested at 18 months had negative results. Among the 81 asymptomatic infants tested at 6 months of age, four (5%) were positive, while all 44 of those tested at 18 months were negative.

## 4. Discussion

The persistence of anti-ZIKV-IgG antibodies and applicability for the diagnosis of ZIKV congenital infection had not yet been established. In this study, we defined severely affected infants as those with presumed ZIKV congenital infection and compared them to a group of asymptomatic ZIKV-exposed infants, expecting that the persistence of anti-ZIKV-IgG antibodies after 1 year of age would be much more frequent in the presumed infected infants. We found neither of the used conventional markers such as nucleic acid testing and IgM useful to discriminate infants infected in intrauterine life with this virus. No infant tested positive for ZIKV-PCR independently of the presence of Zika-like clinical findings. Only 60% of the severely affected infants had detectable anti-ZIKV-IgM antibodies, and all infants, including those presumptively infected and those with detectable anti-ZIKV-IgM, lost anti-ZIKV-IgG antibodies within 15 months of age.

Data from a prospective cohort that extensively tested fetuses/neonates born to infected women [4] identified that RNA detection was far more effective in the cerebrospinal fluid (CSF) (57%), amniotic fluid (42%), and placental (22%) specimens than in the umbilical/neonatal cord blood (17%) or urine (3%). Even among affected fetuses/neonates, this testing was rarely (3.4%) positive in blood. Other studies also showed low figures of ZIKV-RNA (0.3%, 1/295) in multiple specimens [18] and blood [19], agreeing with our findings and suggesting almost zero neonatal ZIKV-RNA positivity. In contrast, in another study, ZIKV-RNA positivity in the serum (35%) and especially in the urine (49%) of 130 infants born to ZIKV-infected mothers tested during the first year of life was reported to be higher [20], with fluctuating results in few infants tested more than once, but post-natal ZIKV infection could not be ruled out. Viral and assay factors likely explain the infrequent neonatal detection rate of ZIKV-RNA in blood in most studies. The ZIKV viral load in adults has been extensively described as low (about 102 RNA copies/mL), variable, and usually brief [21,22,23]. Fetal viremia and viruria have not yet been fully described, making it difficult to predict if fetuses exposed early in pregnancy, usually the most severely affected [2,7], would still have detectable RNA at birth [9]. Additionally, the wide variety of molecular test systems used worldwide, the lack of standardization and the resulting low concordance across different platforms/laboratories, and the overall variable and low sensitivity [24,25] are important limitations of RT-PCR for the diagnosis of ZIKV congenital infection.

Anti-ZIKV-IgM antibodies have been more frequently detected in ZIKV-exposed infants than ZIKV-RNA, ranging from 37% [20] to 43.6% [4] in the blood of non-selected infants. In contrast, one report identified these antibodies in the blood (90.5%) and all 30 CSF samples collected within 2 months of life of microcephalic infants [19]. We detected IgM in about half of the CZS infants and in only two (1.7%) asymptomatic infants. As in other infants with CZS in our study, these two asymptomatic infants were repeatedly positive in subsequent samples obtained over time. Because IgM molecules do not cross the placenta, IgM detection could indicate adequate antibody fetal responses to a true infection. However, due to its cross-reactivity with the IgM of other flaviruses [23], a false-positive result might occur. So, its use as an isolated marker of congenital infection in an asymptomatic infant is likely limited.

It has been already demonstrated that anti-ZIKV-IgG antibodies can be detected in adults from 2 weeks after infection [16,26] and that sustained transplacental IgG transfer does not seem to be impacted by ZIKV placental infection [27,28]. A single time point testing of total and neutralizing anti-ZIKV-IgG serum antibodies in 11 microcephalic infants at a median age of 5 months resulted in 100% positivity [29]. We have found positive results in all ZIKV-exposed infants tested at birth and 3–6 weeks. However, differently from infants with latent congenital infections such as T. gondii, Cytomegalovirus (CMV), and Human Immunodeficiency Virus [10,30], these antibodies did not persist for months after birth. They waned in severely affected infants as rapidly as asymptomatic infants and similarly to other self-limited infections such as dengue [31,32], which is rarely vertically transmitted.

The waning anti-ZIKV-IgG response over time was previously described in pregnant Rhesus macaques [33] and ZIKV-exposed primate infants [34]. For the latter, neutralizing antibodies were detectable by 2–3 months of age, and IgG titers decreased progressively between birth and 3–4 months of age, although they were still detectable before reinoculation with ZIKV at 5 months of age [34]. These findings suggest that the temporary postnatal blood detection of IgG refers to the presence of antibodies transferred from the mother through the placenta instead of the fetal development of the adaptative antiviral immune response. Fetal immune response to microorganisms can be affected by infection timing during pregnancy, fetal immune system ontogeny, and type of infection [35]. Therefore, maternal infection close to the term of pregnancy could not result in a detectable fetal response due to insufficient time. On the other hand, the most severely ZIKV-affected infants, usually secondary to first-trimester maternal infections [2], could not mount a robust adaptative immune response due to the immature fetal immune system. Our study observed the non-persistence of anti-ZIKV-IgG antibodies even in affected infants with positive anti-ZIKV-IgM results and in the subset of infants with a long-term detectable IgM antibody response. It is likely that, among these IgM-positive infants, the B-cell switch for the production of the IgG class antibodies occurred. One could hypothesize that, although present in fetal life, anti-ZIKV-IgG fetal antibodies have decreasing concentrations until birth and the early months of life, which could not be detected by the assay used.

We chose to use a commercially available assay that detected anti-ZIKV-IgG antibodies in 47% of adult blood samples collected within 4 days of disease onset and in 100% of samples obtained between days 31–300 [16]. Thus, its performance under the circumstances with low antibody concentrations might not be ideal. To address the possibility of the low accuracy of this assay, we subsequently tested infants’ sera from 6 and 18 months of age with the NS1 BOB assay, which had been used in a heterogeneous panel of blood from children and adults with a confirmed ZIKV infection and/or other flaviviruses. The sera were tested with a reported sensitivity between 92–95% and specificity between 80.4–95.9% [17]. However, with this assay, much fewer infants had detectable antibodies at 6 months of age. Considering that in utero immunological priming to ZIKV could have occurred in most infected infants, specific serological or cellular assays designed for diagnosing this congenital infection are desirable.

To our knowledge, this is the first study in ZIKV-exposed human infants assessing the prospective longitudinal kinetics from birth to 18 months of age of anti-ZIKV-IgG as a marker of congenital infection. Its design enabled a comparison of laboratory test results between a cohort of asymptomatic ZIKV-exposed infants versus CZS-affected infants for whom other causes of central nervous system abnormalities were excluded [2]. Due to herd immunity, it is highly unlikely that the recruitment of cohorts of this magnitude will be feasible in the future. Nonetheless, there remains interest in enhancing knowledge about the management of ZIKV-exposed infants, as there is ongoing endemic transmission globally [36]. Our study was limited by using a commercially available method validated for detecting antibodies in children and adults with ZIKV-acquired infection and not specially designed to detect low antibody levels in infants. However, it was our objective to test one already available assay that could be applied in routine practice as is done for other congenital infections. Additionally, it was not possible to study T cells response as a means of better understanding the neonatal immunological response.

In conclusion, although the detection in blood of anti-ZIKV-IgM within 6 months of age is markedly different between groups of CZS and asymptomatic infants, it would still not be helpful to diagnose congenital disease in 40% of patients who are probably affected. The persistence of anti-ZIKV-IgG antibodies until 18–24 months of age in this population does not seem to provide additional diagnostic support. Therefore, postnatal ZIKV diagnosis remains a challenge, especially for asymptomatic exposed infants, and it will probably continue to rely on the analysis of multiple specimens collected as early as possible in the neonatal period using a combination of different testing platforms.

## Figures and Tables

**Figure 1 viruses-13-00711-f001:**
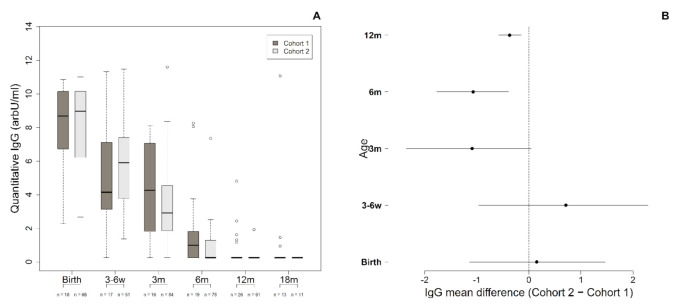
(**A**) A comparison of Anti-ZIKV-IgG levels from birth to 18 months of age in infants born to ZIKV-infected mothers according to the presence of typical CZS (Cohort one) or no potentially ZIKV-related signs (cohort two). (**B**) A forest plot showing IgG mean differences between Cohorts one and two from birth to 12 months of age and their respective 95% credible intervals.

**Figure 2 viruses-13-00711-f002:**
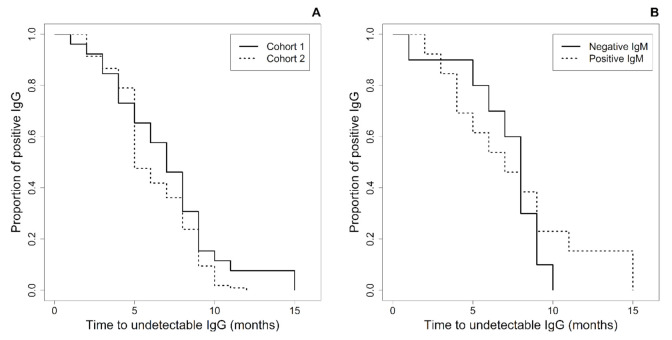
(**A**) Time to undetectable anti-ZIKV-IgG in infants born to ZIKV-infected mothers according to the presence of typical Congenital Zika Syndrome (CZS) (Cohort one) or no potentially ZIKV-related signs (Cohort two). (**B**) Time to undetectable anti-ZIKV-IgG in infants with CZS according to the reactivity of the anti-ZIKV-IgM within 6 months of age.

**Table 1 viruses-13-00711-t001:** The general characteristics of infants from both cohorts.

Characteristics	Cohort One (CZS) (*n* = 30)	Cohort Two (Asymptomatic) (*n* = 123)
Trimester of maternal ZIKV infection		
1st	20 (66.7%)	27 (21.9%)
2nd	1 (3.3%)	72 (58.5%)
3rd	0	23 (18.7%)
No rash	8 (26.7%)	0
Missing	1 (3.3%)	1 (0.9%)
Gestational age		
Term	24 (80.0%)	110 (89.4%)
Preterm	5 (16.7%)	11 (8.7%)
Missing	1 (3.3%)	2 (0.8%)
Birth weight (g)	2694 (623)	3101 (661)
Head Circumference (cm)	28.3 (5.6)	33.4 (4.2)

CZS: Congenital Zika syndrome. ZIKV: Zika virus.

**Table 2 viruses-13-00711-t002:** ZIKV-RNA and anti-ZIKV-IgM in both infants’ groups.

Test	Cohort One (CZS) (*n* = 30)	Cohort Two (Asymptomatic) (*n* = 123)
ZIKV-RNA at birth (blood and/or urine)		
Positive	0	0
Negative	17 (56.7%)	45 (36.6%)
Indeterminate	0	1 (0.8%)
Missing	13 (43.3%)	77 (62.6%)
Anti-ZIKV-IgM within 6 months		
Positive	15 (50.0%)	2 (1.6%)
Negative	10 (33.3%)	113 (91.9%)
Missing	5 (16.7%)	8 (6.5%)

CZS: Congenital Zika syndrome. ZIKV: Zika virus.

## Data Availability

The data that support the findings of this study are available from the corresponding author upon reasonable request.

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
