# Peer review of "Persistence of Anti-ZIKV-IgG over Time Is Not a Useful Congenital Infection Marker in Infants Born to ZIKV-Infected Mothers: The NATZIG Cohort"

_viruses, 2021, doi:10.3390/v13040711_

Round 1

Reviewer 1 Report

In the study by Coutinho et al the authors try to determine whether persistence of anti ZIKV IgG and presence of IgM in infants born with CZS vs asymptomatic were different. Overall, the study is well conducted and presented. I have a few comments that the authors need to address

  1. The authors correctly point out that CZS correlates with the trimester of infection. The early infection in fetus while more pathogenic is also less likely to illicit an immune response in the fetus due to the immature immune system. The authors should stratify the data based on trimester of infection (or perhaps month of infection) and see if a correlation is seen in the time of infection and fetus derived immune response. For the 40% of the CZS subjects that did not have IgMs was the infection at an earlier time point. Also was there a correlation between severity of disease and presence or absence of IgM.
  2. Have the authors considered looking for T cells response in these subjects. It is possible that memory T cell responses persist even when the IgG responses wane; provided these antibodies were not passively transferred from the mother to the fetus.
  3. While the study in itself is interesting it is unclear whether the objective of the study was to develop a diagnostic assay for congenital zika infection. As these analyses are done after birth, there are other established clinical criteria for CZS that can diagnose developmental disorder. How would the detection of IgMs or IgGs up to 18 months after birth change or compliment the clinical diagnosis of CZS.

Reviewer 2 Report

This manuscript is very well written with in-depth knowledge. It can be accepted in the present form. 

Reviewer 3 Report

In this manuscript, authors have compared the presence of anti-ZIKV-IgM and the persistence of anti-ZIKV-IgG antibodies in two cohorts of infants born to ZIKV-infected mothers. Authors claimed that the persistence of IgG is not a useful marker for discriminating congenital infection defined as CZS among infants exposed to maternal ZIKV infection. This finding could be important to the field, however the way it is described is very confusing. For example: the abstract, should only contain the most relevant and summarized results. Result sections, it is very hard to understand by just reading, authors need to systematically describe their findings. Table 2, contain information that are not described in the text such as: “Missing”. The confidence interval (CI) should be summarized in the table 2 rather than in the text. I strongly encourage the authors to describe systematically the results and abstract. Additionally, it is highly recommended to the authors to obtain an assistance of a native English speaker to adjust the style required.

Round 2

Reviewer 1 Report

The authors have addressed all the comments.